# Climate model-informed deep learning of global soil moisture distribution

Klaus Klingmüller[1] and Jos Lelieveld[1,2]

[1]Max Planck Institute for Chemistry, Hahn-Meitner-Weg 1, 55128 Mainz, Germany
[2]The Cyprus Institute, P.O. Box 27456, 1645 Nicosia, Cyprus

**Correspondence:** Klaus Klingmüller (k.klingmueller@mpic.de)

**Abstract.** We present a deep neural network (DNN) that produces accurate predictions of observed surface soil moisture, applying meteorological data from a climate model. The network was trained on daily satellite retrievals of soil moisture from the European Space Agency (ESA) Climate Change Initiative (CCI). The predictors precipitation, temperature and humidity were simulated with the ECHAM/MESSy atmospheric chemistry-climate model (EMAC). Our evaluation shows that predictions of the trained DNN are highly correlated with the observations, both, spatially and temporally, and free of bias. This offers an alternative for parametrisation schemes in climate models, especially in simulations that use, but may not focus on soil moisture, which we illustrate with the threshold wind speed for mineral dust emissions. Moreover, the DNN can provide proxies for missing values in satellite observations to produce realistic, comprehensive, high resolution global datasets. As the approach presented here could be similarly used for other variables and observations, the study is a proof of concept for basic but expedient machine learning techniques in climate modelling, which may motivate additional applications.

## 1 Introduction

Since decades, global climate and atmospheric-chemistry models rely on supercomputers with a traditional cluster architecture, utilising many powerful compute nodes in parallel. The individual nodes are typically very potent by themselves, using general purpose central processing unit (CPU) cores with large memory. They are required to serve the needs of diverse algorithms representing many different physical and chemical processes. The implementations of these algorithms quite often have a source code legacy with a history of multiple decades. This creates a dependency on the system type which may limit progress in model studies once the CPU performance increase slows down, typical for the recent past. In fact, due to challenges in the continued miniaturisation of semiconductors, such a slowdown is ongoing, which motivates the search for other performance gains (Leiserson et al., 2020).

Progress has been made to accelerate weather, climate and atmospheric-chemistry models with general purpose graphics processing units (GPUs) (Yashiro et al., 2016; Alvanos and Christoudias, 2017; Sun et al., 2018; Fuhrer et al., 2018; Müller et al., 2019), but a wider utilisation is pending. Other high performance computing applications such as lattice quantum chromodynamics with typically smaller codebase and less legacy code swiftly exploited the computational resources of GPUs (Egri et al., 2007; Clark et al., 2010). A discipline which not only uses the full potential of GPUs but also builds its present success

on the advent of GPU computing, is machine learning, particularly deep learning (Krizhevsky et al., 2012). Strong commercial interest even boosted the development of specialised machine learning hard- and software (e.g. Abadi et al., 2015; Jouppi et al., 2017; Markidis et al., 2018). Atmospheric and climate models can benefit from this development by making use of the computational capabilities of the new hardware to accelerate existing algorithms (Hatfield et al., 2019), or by applying machine learning techniques to complement existing, physical process-based and in particular empirical parametrisations (Chevallier et al., 2000).

Given the success of deep learning in many different disciplines (Schmidhuber, 2015; LeCun et al., 2015; Silver et al., 2016) it seems likely that atmospheric modelling and climate science in general can benefit from this methodology, not only in terms of performance gains, and promising results have been presented lately (e.g. Kadow et al., 2020; Chandra et al., 2021). However, reservations about machine learning exist. In contrast to physical models, where the simulation result is deduced from the laws of physics and physical parameters, trained models often represent black boxes where the rules by which they compute the output tend to be non-transparent and cannot easily be modified to represent varied physical conditions. On the other hand, complex conventional, phenomenologically derived parametrisations at times also lack a clear physical meaning and their parameters are often tuned to obtain realistic results, which essentially is a non-systematic form of training. Moreover, methods for interpreting machine learning models are emerging (Montavon et al., 2018; Kohoutová et al., 2020). Aside from that, depending on the application, it may be irrelevant whether a process is implemented as a black box or not if the scientific focus is on other processes. For instance, the present study was motivated by the need for reliable soil moisture data to develop a mineral dust emission scheme.

Soil moisture near the surface has a significant impact on the emissions of mineral dust (Klingmüller et al., 2016) and it is generally of great importance for weather and climate. It has been identified by the Global Observing System for Climate (GCOS) as an essential climate variable (ECV, Bojinski et al., 2014) and, for example, soil moisture can greatly impact mesoscale convective systems (Klein and Taylor, 2020). While detailed parametrisations of soil moisture exist (e.g., Ekici et al., 2014), many models, such as the ECHAM/MESSy atmospheric chemistry-climate model (EMAC) (Jöckel et al., 2006) still use relatively simple models.

On the other hand, because moisture at the surface affects the dielectric properties of the soil, it can be well retrieved remotely using microwaves, and an extensive global daily dataset covering the past four decades is provided by the European Space Agency (ESA) Climate Change Initiative (CCI) (Dorigo et al., 2017; Gruber et al., 2017, 2019).

To make use of the ESA CCI surface soil moisture data in climate models, it cannot be imported directly, because the daily subsets have substantial gaps depending on the local retrieval conditions at overpass time. Moreover, merely importing observations would limit model applications to hindcasting. Therefore we pursued an alternative approach and use the satellite data for supervised training of a deep neural network (DNN) to predict soil moisture based on modelled meteorological data. In doing so, we explored the potential of machine learning to complement physical models using an introductory example, which demonstrates the useful results which can be achieved with limited technical effort, and which might instigate further applications.

This article is structured as follows: the datasets used are presented in section 2, the DNN is introduced and evaluated in section 3, applications of the DNN are discussed in section 4 before we draw conclusions in section 5.

## 2 Data

We used results from a 10 year simulation with the ECHAM/MESSy atmospheric chemistry-climate model (EMAC) (Jöckel et al., 2006) version 2.52, covering the years 2006 to 2015. The exact configuration is described by Klingmüller et al. (2020). Horizontally, the setup employs a Gaussian T63 grid with approximately 1.9° spacing. EMAC assimilates observational data by nudging temperature, vorticity and divergence above the boundary layer to meteorological reanalysis data of the European Centre for Medium-Range Weather Forecasts (ECMWF) and by using the sea surface temperature from the same dataset.

The EMAC output variables we considered are precipitation, surface temperature and humidity, which are preprocessed to daily average values. In addition, we made use of the static ecosystem rooting depth map of the online dust emission scheme, originally from Schenk and Jackson (2009).

The EMAC data was complemented by daily volumetric soil moisture (i.e. the ratio of water relative to soil volume) observations from the ESA CCI Soil Moisture Product Release v04.5 (Dorigo et al., 2017; Gruber et al., 2017, 2019). This dataset is representative for the soil moisture in the topmost few centimetres of soil (down to about 5 cm). We used the dataset combining retrievals from active and passive spaceborne microwave instruments, which we aggregated from the original spatial grid with 0.25° spacing to the Gaussian T63 grid of the EMAC results.

We subdivided the 10 year period covered by the EMAC simulation into a training period of 8 years from 2006 to 2013 and a test period of 2 years from 2014 to 2015. This choice maximises the length of the training period while keeping more than one year for testing, allowing to identify interannual variations during the testing period. Moreover, using training and testing periods in chronological order, all predictions in the testing period represent forecasts. The test period was exclusively used to evaluate the DNN after training. Every third year of the training period (2008 and 2011) was used for validating and monitoring the training procedure.

## 3 DNN model

The basic concept of our approach is to relate the observed soil moisture to relevant quantities modelled by the global climate model. We applied a simple DNN architecture which operates on one grid cell at a time. As a consequence, the DNN can easily be integrated as a submodel in global climate models such as EMAC. The meteorological variables used as predictors are selected based on availability in the climate simulation output and physical relevance: the rain rate represents the source of soil moisture, whereas surface temperature and atmospheric humidity control soil drying. To account for cumulative effects, in addition to the actual daily mean rain rate, surface temperature and specific humidity, we provided the DNN with the corresponding values lagged by 1 day, the mean of values with 2 to 3 days lag, and the mean of 4 to 7 days lagged values. According to preliminary tests, considering longer lags did not yield a significant improvement, but it might become relevant if

the overall performance can be enhanced in future versions. To account for regional characteristics such as soil properties, the DNN uses longitude and latitude, encoded in the triple sin(lon), cos(lon), sin(lat) as well as the local rooting depth. Likewise, to account for seasonality, for example by vegetation variations, we supplied the DNN with the time of year encoded in $\sin(2\pi t/\mathrm{a})$ and $\cos(2\pi t/\mathrm{a})$, where $t/\mathrm{a}$ is the time in years. In total this amounts to 18 input variables which had to be mapped to one output variable, the surface soil moisture.

For that purpose, we employed a generic DNN of linearly stacked densely connected layers as illustrated in Fig. 1. Four hidden layers of 512 units with rectified linear activation are followed by the output layer with a single unit and linear activation. There is no strict formula for the number of layers and units per layer, but too small DNNs are not capable of learning complex rules, whereas too large DNNs are more at risk of overfitting and require more computational resources. Our values are a compromise that is proving to work well, but systematic optimisation could possibly yield better results. To generalise the DNN and prevent overfitting during training, we regularised by applying a 10 % dropout rate to the hidden layers (Hinton et al., 2012) and stopped the training process as soon as the validation loss was no longer improving. Before training, all input and output variables were normalised independently to have a mean of 0 and a standard deviation of 1. Accordingly, before using the DNN for predictions, the same transformations had to be applied to the input data and an inverse transformation to the output.

We implemented the DNN and performed the training and inference using the TensorFlow library (Abadi et al., 2015) with the Keras interface (Chollet et al., 2015, 2017) for R (R Core Team, 2019). The training took about 40 minutes on an Nvidia Tesla V100 GPU accelerator. The computationally much less demanding predictions were evaluated on common desktop hardware.

To assess the overall predictive power of the DNN, we compared the volumetric soil moisture calculations with the corresponding observations for all grid cells and days during the test period (2014 to 2015) where retrievals are available. Figure 2 shows the scatter of the 1247041 observation-prediction pairs. It demonstrates a remarkably high quality of the predictions, which are strongly correlated with the observations, with a Pearson correlation coefficient of 0.92, and do not show any bias, resulting in a small root-mean-square error of 0.033 and a mean absolute error of 0.024, being an order of magnitude smaller than the average data values. Note that the Gaussian grid has more grid cells per area at higher latitudes, giving those regions more weight in this comparison, but the number of relevant grid cells in polar regions is small.

While Fig. 2 combines the effect of the spatial and temporal variability, Fig. 3 shows the spatial correlation for each day separately, using all grid cells with observations during that day. The correlation coefficient attains high values around 0.92 throughout the test period and rarely drops below 0.9. Considering that the training takes place before the test period, it is noteworthy that there is no significant decline of the correlation over the two years. Essentially, at any time the DNN yields a realistic representation over the globe, predicting low soil moisture in arid regions and high soil moistures in wet areas.

Equally important is a realistic representation of the temporal variation for each grid cell. Due to the strong temporal variability of weather and in particular precipitation, this is more challenging and we do not expect equally high correlation coefficients as for the spatial correlation. Nevertheless, Fig. 4 shows that the temporal correlation coefficients are globally high with a mean of almost 0.7, which appears to be a good performance when compared with state of the art model data (Dorigo et al., 2017).

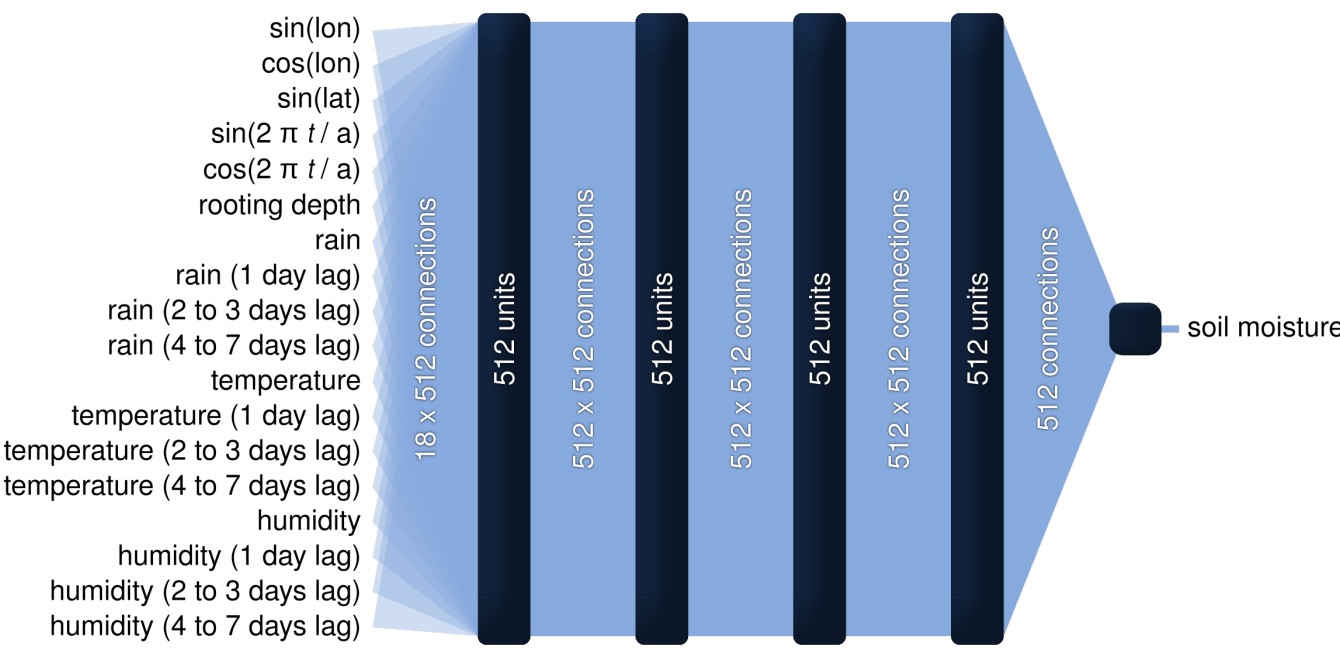

**Figure 1.** DNN architecture and hyperparameters. The input variables on the left are processed by four fully connected hidden layers of 512 units each to yield the soil moisture on the right. All four hidden layers use a rectified linear unit function activation and during the training a dropout rate of 10 % to avoid overfitting. The output layer with only one unit applies a linear activation function to obtain the final result. The DNN operates on normalised variables with standard deviation 1 and mean 0.

The correlation coefficient is larger than 0.5 in almost all grid cells except those representing extremely dry soils of the Sahara, Rub' al Khali, Taklamakan and Gobi deserts. Related to this, a slight overestimation by the DNN towards the lower end of the soil moisture range can be identified in Fig. 2. Training and evaluation of the DNN are challenging in this range because satellite retrievals are both sparse and uncertain (see, e.g., Fig. S1 in the supplement), therefore, depending on the application, some additional efforts that focus on these regions might be required. For dust emissions, the soil moisture in these regions is

too low to significantly influence the results, while it is more relevant in semi-arid regions, where observations and predictions are more reliable (Fig. 4 and Fig. S2 in the supplement).

Considering the grid cell centred at 49.4°N 7.5°E, the upper panel of Fig. 5 exemplifies how closely the predicted volumetric soil moisture time series resembles the observations. There is no bias in the predictions and the seasonal cycle is well represented. Moreover, the short-term variations of the predictions show a clear similarity to those of the observations. Both have a

comparable amplitude and frequency, and characteristic features in the observed time series are reproduced by the predictions, e.g., in July 2014, October 2014, December 2014/January 2015 or March 2015 (Fig. S3 in the supplement). These features

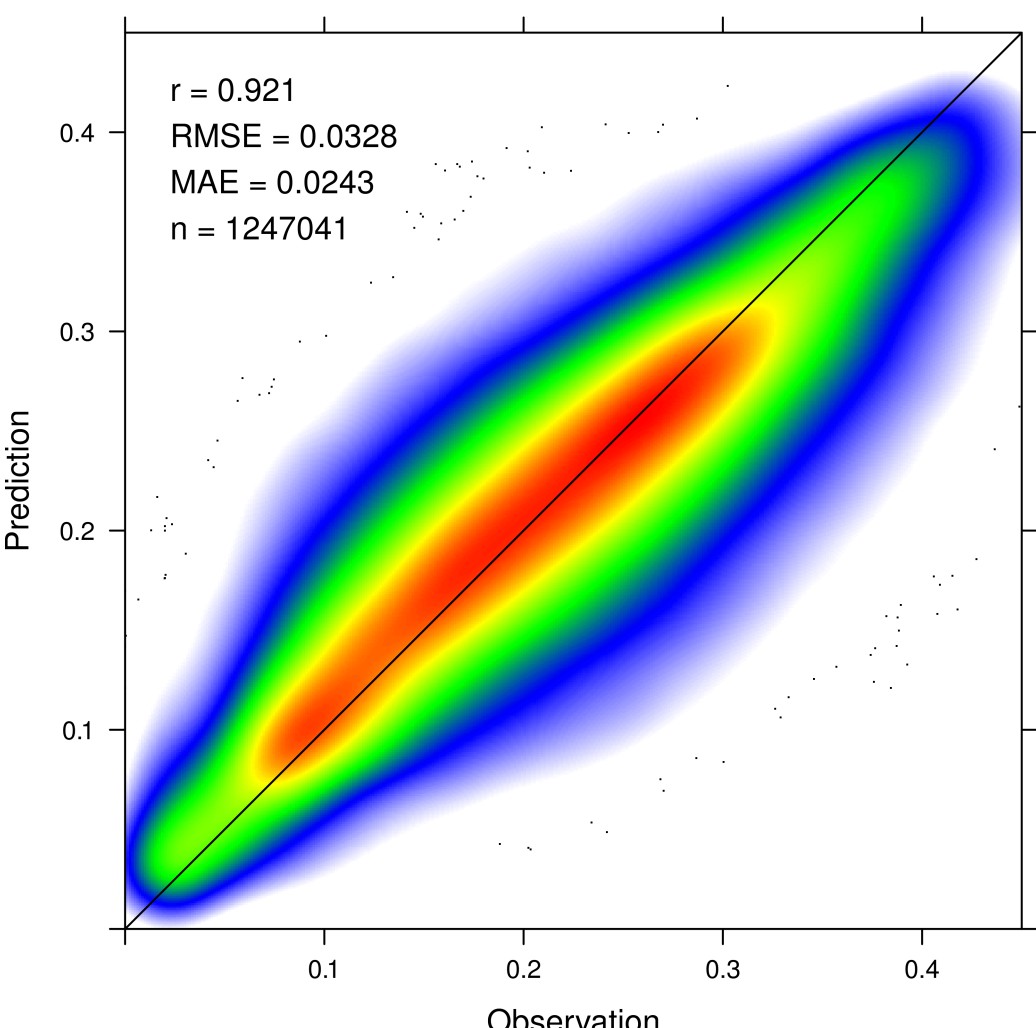

**Figure 2.** Comparison of observed and predicted volumetric surface soil moisture considering all daily grid-cell values available globally during the years 2014 and 2015. The colours represent the density distribution of the scatter, outliers are represented by dots. Pearson correlation coefficient $r$, root-mean-square error (RMSE), mean absolute error (MAE) and the number of data points $n$ are provided in the upper left corner.

## Spatial correlation

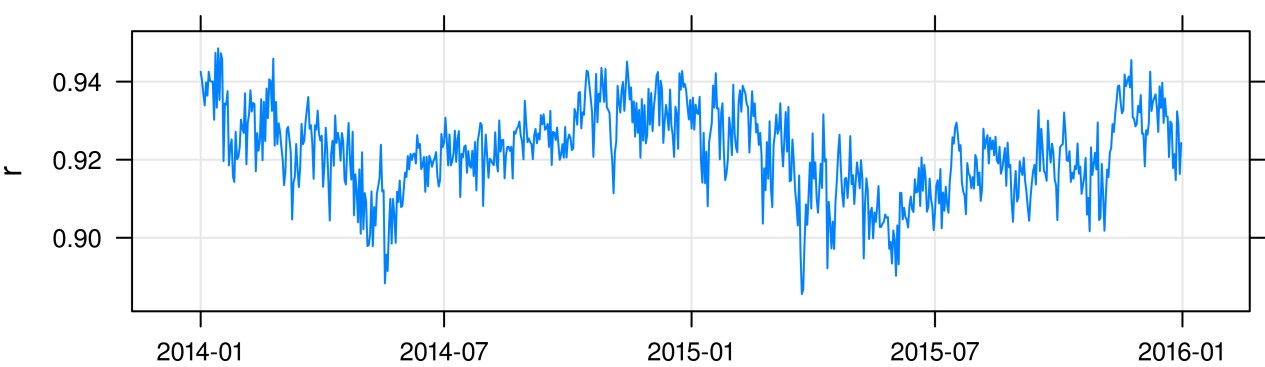

**Figure 3.** Spatial correlation of predicted and observed volumetric soil moisture throughout the test period.

## Temporal correlation

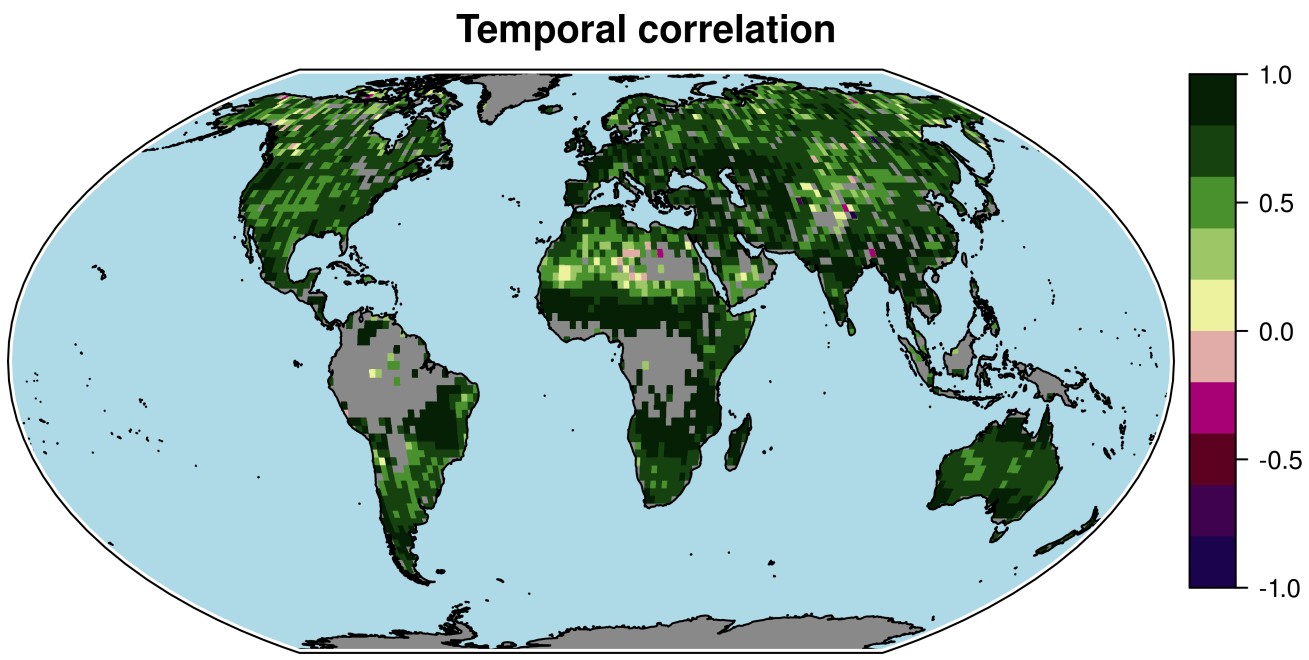

**Figure 4.** Correlation coefficient of the observed and predicted volumetric soil moisture time series in the individual grid cells during the test years 2014 and 2015. In the grey regions no observations are available during the this period.

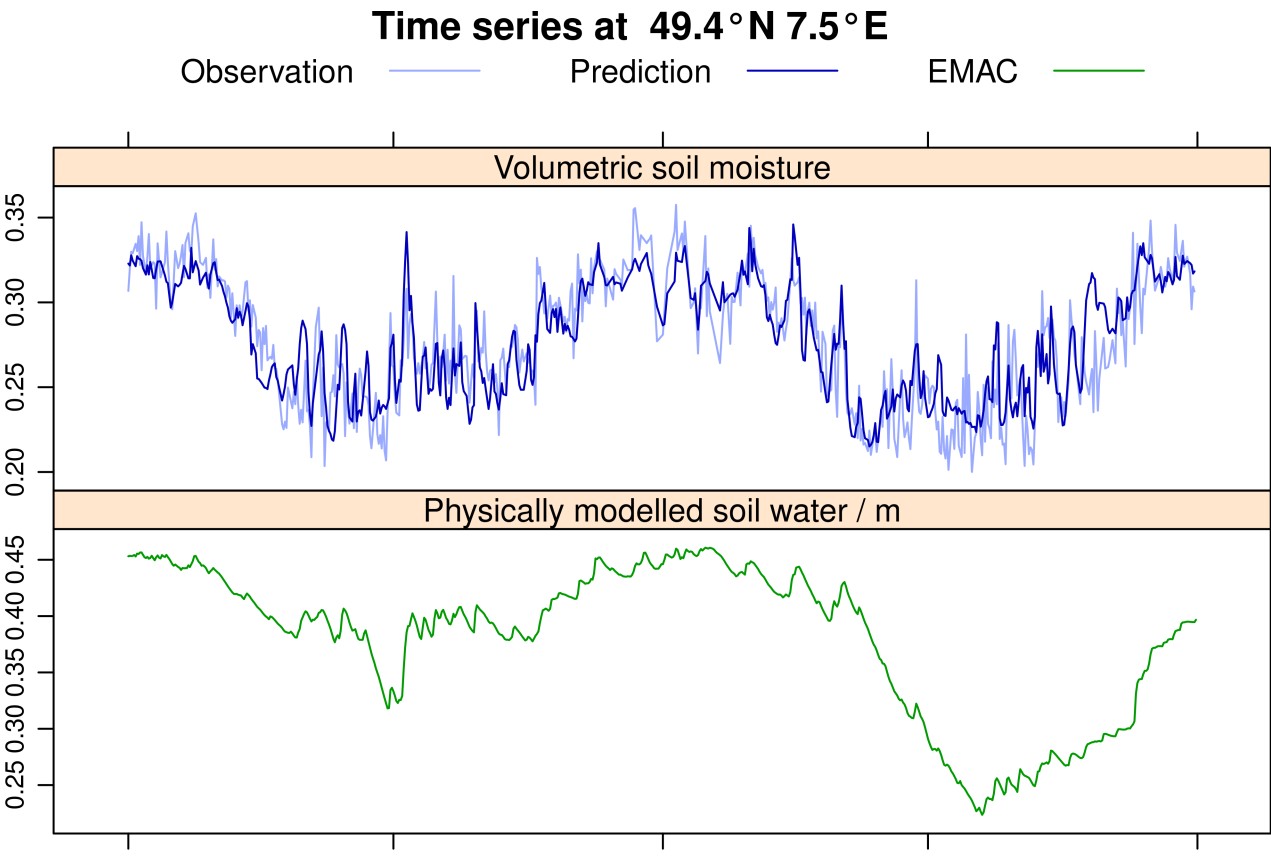

**Figure 5.** Time series of the observed and predicted daily volumetric soil moisture values and the EMAC soil water in the grid cell centred at 49.4°N 7.5°E in Europe.

occur irregularly and are not repeated year after year, which demonstrates that the DNN did not simply learn one representative climatology but utilises information from the meteorological data provided by the climate model. We reiterate that the nudged EMAC meteorology is close but not identical to the real atmosphere, hence small deviations are expected.

For the realistic representation of the temporal soil moisture variation in each grid cell, the meteorological predictors are essential. The climatological model obtained by training the DNN omitting the meteorological predictors still reaches an overall correlation coefficient of close to 0.9, but the global mean of temporal correlation coefficient drops substantially from close to 0.7 to below 0.5 (Fig. S4 in the supplement). In addition to the interannual variations, in particular the variability on time scales shorter than 1 month is lost by ignoring the meteorology (Fig. S5 in the supplement). The relevance of the individual

meteorological predictors for the temporal variations differs regionally. For example in Europe, the DNN predictions are sensitive to all meteorological predictors, but in arid and semi-arid regions of the Middle East they are not sensitive to reducing the rain (Fig. S6 to S8 in the supplement). Here, all variability is inferred from temperature and humidity. This does not

necessarily imply rain to be the least important driver of soil moisture variability in the region, rather it is the least reliable predictor due to the uncertainty in modelled precipitation.

The meteorological predictors are also required to make the DNN sensitive to climatic changes. Increasing temperature or decreasing humidity promotes soil drying, whereas increasing rain enhances the soil moisture, which is reproduced by the DNN (Fig. S6 to S8 in the supplement). Therefore, if climatic changes are represented in the meteorological predictors, the DNN soil moisture will respond to theses changes. The response might be limited because in the present configuration the DNN can substitute meteorological information with the knowledge of season and location. Consequently, for applications in

long-term climate projections, training and evaluation on longer time scales and possibly refinements of the DNN architecture are advisable.

    The spatial coordinate predictors are crucial for a realistic spatial variation of the soil moisture. They are the parameters encoding information on the wide range of surface properties including soil and vegetation types, and substantially improve the DNN. Less relevant are the seasonal predictors, but if provided, the DNN makes use of them so that the DNN predictions

are sensitive to variations of the time of year (Figs. S6 to S8 in the supplement) and the temporal correlation is improved (Fig. S4 in the supplement).

## 4   Applications

The DNN operates on a single grid cell at a time and therefore can easily be incorporated in climate models, for example in the submodel core layer of EMAC. This provides realistic soil moisture values to other submodels, which can optionally replace

the current parametrisation of soil hydrology. For instance, this could be advantageous for mineral dust emission schemes which should account for reduced emissions from wet soils but so far have been limited by the inadequate representation of soil moisture in the topmost surface layer by the physically modelled soil water (Klingmüller et al., 2018).

    The lower panel of Fig. 5 shows the time series of the physically modelled EMAC soil water. Evidently, this time series is largely unrelated to the observed surface soil moisture in the upper panel: the short-term variability is smaller whereas the

170 long-term variability is much larger, showing a strong decline during summer 2015 which is not present in the observations. Regardless of the question whether this decline reflects a true decline of the water content in deeper soil layers or whether it is only a model artefact, it is obviously impossible to map the EMAC soil water to a realistic representation of the observed surface soil moisture in the panel above. However, it is the latter which is required for parametrisations such as the mineral dust emission scheme. Because the DNN presented here fulfils this requirement, we propose to use it as improvement over

175 the EMAC computed soil water and as viable alternative to more sophisticated physical soil moisture models. The algorithm cannot replace physically-based soil moisture representations for first-principle process studies, but is an accurate substitute for parametrisations that depend on limited empirical information.

    Mineral dust emissions are predominantly caused by saltation bombardment where saltating particles on impact with the surface eject finer dust sediments or disintegrate to finer particles themselves. To activate and sustain a horizontal flux of

180 saltating particles, the surface friction velocity of the air has to exceed a threshold which depends on the soil properties.

Soil moisture increases this threshold, thereby reducing the dust emissions. We studied this effect using the parametrisation presented by Fécan et al. (1999),

$$\frac{u_{*\mathrm{t}}}{u_{*\mathrm{td}}} = \sqrt{1 + 1.21(w - (0.0014\phi_{\mathrm{clay}}^2 + 0.17\phi_{\mathrm{clay}}))^{0.68}}, \tag{1}$$

where $u_{*\mathrm{t}}$ is the threshold surface friction velocity, $u_{*\mathrm{td}}$ the corresponding threshold for dry soil, $w$ the gravimetric soil moisture in percent and $\phi_{\mathrm{clay}}$ the soil clay fraction in percent. The equation is applied if the soil moisture exceeds $0.0014\phi_{\mathrm{clay}}^2 + 0.17\phi_{\mathrm{clay}}$, representing the minimum soil moisture required to induce an increase in the threshold. Like Astitha et al. (2012), we combined this soil moisture dependency with the threshold surface friction velocity parametrisation of Marticorena and Bergametti (1995), applied to a saltation particle diameter of 60 μm. The full equation for the threshold surface friction velocity is reproduced in appendix A. We evaluated the equation for air density $\rho_{\mathrm{air}} = 1.2\,\mathrm{kg\,m^{-3}}$ using the clay fraction distribution from Shangguan et al. (2014). To convert the volumetric to gravimetric soil moisture we assumed a soil bulk density of $1600\,\mathrm{kg\,m^{-3}}$. For comparison, we converted the EMAC soil water from water volume per area to gravimetric soil moisture using the same bulk density and assuming the water to be evenly distributed over the soil column defined by the rooting depth.

We focused our analysis on Mesopotamia and the Arabian Peninsula where a significant correlation of soil moisture and dust emissions was reported (Klingmüller et al., 2016), and considered the regional average of the threshold surface friction velocity over the territory of Iraq, Israel, Jordan, Kuwait, Lebanon, Oman, Qatar, Saudi Arabia, State of Palestine, Syria, United Arab Emirates and Yemen. The threshold surface friction velocity during the test period is shown in Fig. 6 calculated using the observed, predicted and EMAC-calculated soil moisture. The results based on the observed and predicted soil moisture show good agreement and a strong seasonal cycle. During summer, the threshold calculated based on the DNN predictions tends to be slightly higher than that based on the observed values indicating an effect of the aforementioned challenges in hyper arid regions. Nevertheless, the difference between both results is much smaller than the interannual variations and the DNN result is in the range of the short term variations of the observation based result which result from the varying retrieval coverage. In contrast, the result based on the EMAC soil water has little variability, and is therefore inconsistent with the other two results, irrespective of the precise conversion factor used to obtain the gravimetric surface soil moisture. We conclude that the value of the EMAC soil water, which represents the total water including moisture in deeper soil layers, is limited in this context, but so far it has been the only estimate available for EMAC simulations.

Fig. 6 additionally shows results from a recent dataset of Pu et al. (2020) who have retrieved a climatological monthly global distribution of the threshold in terms of the wind speed at 10 m altitude based on satellite and reanalysis data. Two versions of the data represent different assumptions used to identify dust events based on the dust optical depth (DOD). The seasonal cycle is similar to that of the predicted threshold surface friction velocity with a comparable relative amplitude and a similar but slightly shifted phase. Assuming a logarithmic wind profile, the predicted surface friction velocity threshold can be converted to the corresponding 10 m wind speed $u_{10\mathrm{m,t}} = u_{*\mathrm{t}} \ln(10\,\mathrm{m}/z_0)/\kappa$, where $\kappa \approx 0.4$ is the Von Kármán constant and $z_0$ the surface roughness. Consistent with Astitha et al. (2012) we used the surface roughness $z_0 = 0.0001\,\mathrm{m}$. According to the logarithmic profile, the threshold that Eq. (A1) yields for dry soils, i.e., the minimal threshold $u_{*\mathrm{td}} = 0.26\,\mathrm{m\,s^{-1}}$, corresponds to the 10 m wind speed $u_{10\mathrm{m,td}} = 7.5\,\mathrm{m\,s^{-1}}$. This value is higher than most of the climatological values, however, the latter

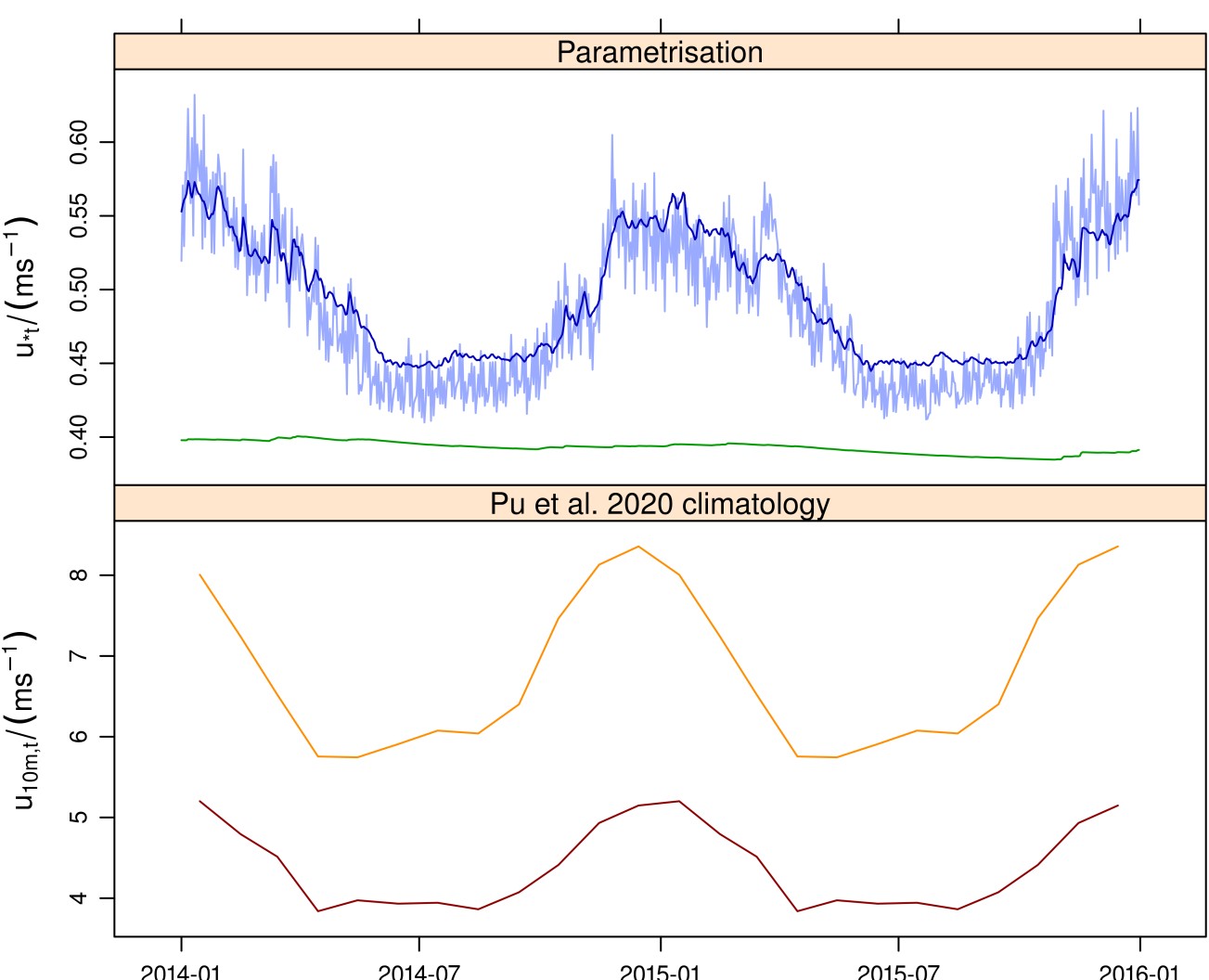

**Figure 6.** The surface friction velocity threshold above which dust is emitted, averaged over the Arabian Peninsula and Mesopotamia. The surface friction velocity threshold $u_{*\mathrm{t}}$ is computed using the parametrisation provided in the appendix A. The threshold in terms of the wind speed at 10 m altitude $u_{10\mathrm{m,t}}$ represents the climatological data set retrieved by Pu et al. (2020).

were derived based on 6-hourly wind speeds, whereas the parametrisation we used is meant for and applied to instantaneous surface friction velocities which vary on shorter time-scales (only limited by the model time step, e.g., 12 min in our T63 simulation and even shorter for higher spatial resolutions) and reach higher peak values. Therefore the threshold definitions differ, not allowing direct comparisons of the absolute values. Additionally, because the retrieval of the climatology does not account for dust transport, it may regionally underestimate the threshold.

The substantial variations of the threshold surface friction velocity obtained based on the observed soil moisture emphasize the relevance of soil moisture for dust emissions. The soil moisture predicted by the DNN is sufficiently realistic to reproduce these variations and to account for this important effect in global climate model simulations.

In addition to the incorporation in global climate models, another application of the DNN is the reprocessing of remote sensing data. Based on meteorological input data, the DNN predicts the global daily soil moisture distribution consistent with the observations. In contrast to the observational datasets that have substantial gaps in regions and time periods where conditions do not allow retrievals, the meteorological input data does not have any missing values, and consequently the same applies to the predicted soil moisture. The latter can therefore also be used to consistently fill the gaps in the observations to obtain a complete daily global soil moisture dataset. Figure 7 shows the global distribution of the observed and predicted soil moisture on two example days from the training period, one during winter on the northern hemisphere (15. January 2012), the other during summer (15. July 2012). Observations outside the training period can be processed as well (Fig. S9 in the supplement) but to obtain optimal results the training should include the period of interest. Regardless of the extensive regions without observations, the prediction yields global values within a reasonable range, closely resembling the observations where available. Note that in regions or seasons where no or only few observations are available throughout the training period, the predictions have to be interpreted with caution. This applies to the rain forests, central deserts and regions permanently or seasonally frozen or covered by snow. In the complete soil moisture distribution the seasonal variations become apparent, promoting the DNN predictions for use in further studies such as trend analysis. Moreover, in contrast to the incomplete observations, the optimized predictions can straightforwardly be assimilated into climate models.

## 5 Conclusions

We have presented a machine learning model which relates soil moisture to meteorological conditions. Informed by a climate model, this DNN is able to accurately predict satellite based surface soil moisture observations, as demonstrated by our evaluation. Using the example of the threshold wind speed for mineral dust emissions we showed that the DNN predictions can be used for improved representations of surface soil moisture-dependent processes within climate models.

The DNN in its present form should be regarded as a proof of concept, and there is room for improvement. The current DNN architecture, the simple stack of several densely connected layers, is very generic. While it is generally quite powerful, it is not tailored to our specific application and other concepts might be considered as well. Convolutional neural networks could exploit the spatial relationship of neighbouring grid cells and recurrent neural networks might more optimally account for the causal relationship of the soil moisture at successive days including long-term accumulative effects. The causal relation is partly

# Volumetric soil moisture

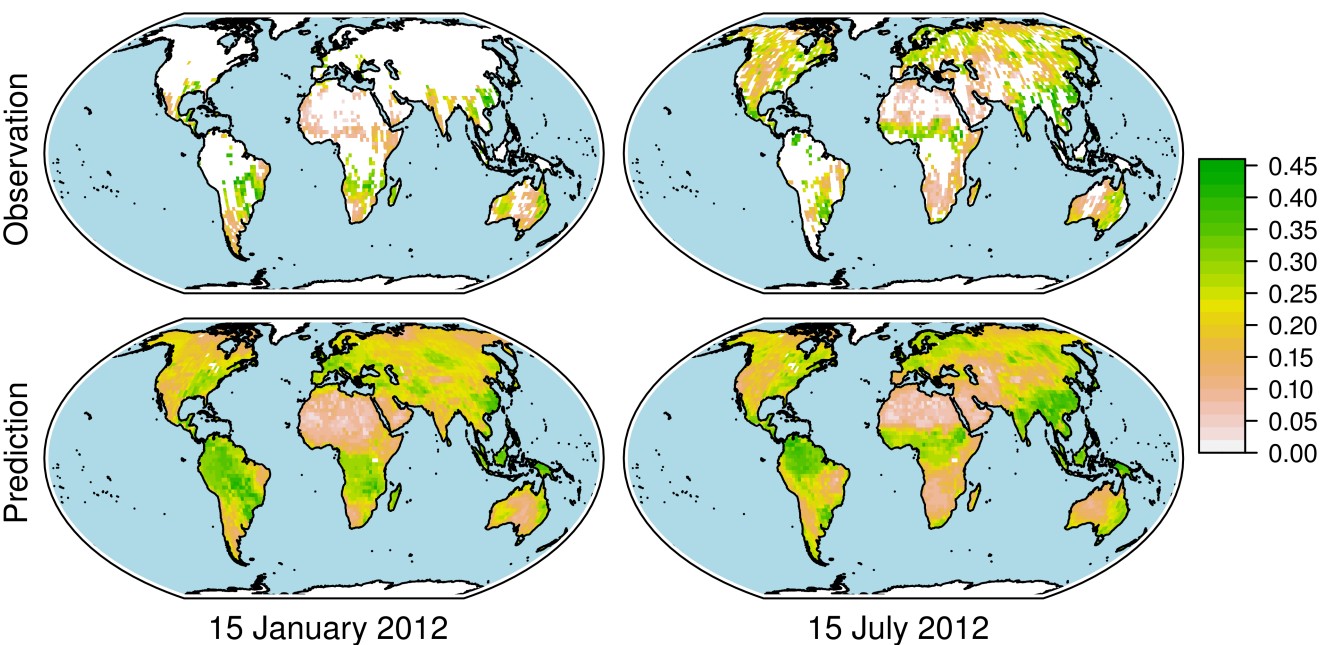

**Figure 7.** Global distribution of the observed and predicted volumetric soil moisture on a northern hemisphere winter (15. January 2012) and summer (15 July 2012) day in the training period.

addressed in our implementation by the consideration of lagged meteorological variables, representing a temporal convolution. However, the prediction is not informed about the conditions prior to one week in the past. This apparently works well for the surface soil moisture, but is probably not sufficient for additional applications, such as those that require information about the moisture in deeper soil layers. The hyperparameters of the DNN, including the number of layers, the number of units per layer and the selection of input variables, are chosen to be appropriate for the problem, but have not been systematically optimised. We conclude that there are various pathways for future developments that may enhance the DNN performance. Nevertheless, the present DNN setup can already be beneficial for applications such as online mineral dust emission schemes in climate models. Therefore, the trained DNN will be implemented as an EMAC submodel using the MESSy interface.

The CPU time required for the inferencing using the trained DNN is negligible compared to the total computational demand of a global climate model. Other applications of DNNs within climate models may be more demanding, in particular if they process three-dimensional data instead of only two-dimensional surface data and if they have to be called more often than once per day. In this case, GPUs or specialised inferencing hardware can be employed to evaluate the trained model, which by design efficiently utilises such accelerators. In this way, climate models can benefit from GPUs that are available in many supercomputers without the need for porting complex algorithms to GPUs, and from the rapid development of machine learning hardware.

For hindcasting applications, an alternative to implementing the trained soil moisture DNN into climate models is to import the consistent and comprehensive global soil moisture prediction from the DNN at runtime. Because such a dataset is also of
265 general use, it appears to be promising to repeat the procedure presented here with high resolution meteorological reanalysis data.

Overall, our example demonstrates that machine learning models informed by data from traditional, physical process-based climate models can perform well in learning and predicting observational data. In return, they can complement process parametrisations in climate models, especially when the parametrisations rely on limited empirical data. Furthermore, they
may help climate models to efficiently utilise recent hardware architectures.

*Code and data availability.* The ECHAM climate model is available to the scientific community under the MPI-M Software License Agreement (https://mpimet.mpg.de/en/science/modeling-with-icon/code-availability, last access: 11 November 2020, MPI-M, 2020). The Modular Earth Submodel System (MESSy) is continuously further developed and applied by a consortium of institutions. The usage of MESSy and access to the source code are licensed to all affiliates of institutions which are members of the MESSy Consortium. Institutions can become a
275 member of the MESSy Consortium by signing the MESSy Memorandum of Understanding. More information can be found on the MESSy Consortium Website (https://www.messy-interface.org, last access: 11 November 2020, MESSy, 2020). The data and DNN parameters used in this study are available at https://edmond.mpdl.mpg.de/imeji/collection/eLt_AnQ98XFaaznl (Klingmüller, 2021).

## Appendix A: Surface friction velocity threshold for dust emissions

The full equation for the threshold surface friction velocity $u_{*t}$ used in section 4 is

$$
\begin{aligned}
u_{*t} =& 0.129 \sqrt{\frac{D_p}{\rho_{\text{air}}} \left( \rho_p g + \frac{0.006 \text{g} \sqrt{\text{cm}/\text{s}^2}}{D_p^{5/2}} \right)} \\
&\times \begin{cases} \frac{1}{\sqrt{1.928 B^{0.092} - 1}} & B < 10 \\ (1 - 0.0858 e^{-0.0617(B-10)}) & B \geq 10 \end{cases} \\
&\times \left( 1 - \frac{\ln \frac{z_o}{z_{\text{os}}}}{\ln(0.35 \left( \frac{10 \text{cm}}{z_{\text{os}}} \right)^{0.8})} \right)^{-1} \\
&\times \sqrt{1 + 1.21 \max(0, \left( w - (0.0014 \phi_{\text{clay}}^2 + 0.17 \phi_{\text{clay}}) \right))^{0.68}},
\end{aligned}
\tag{A1}
$$

where

$D_{\mathrm{p}} = 60 \ \mu m$             saltation particle diameter

$\rho_{\mathrm{air}}$             air density

$\rho_{\mathrm{p}} = 2.65 \ \mathrm{g/cm}^3$             particle density

$g = 9.80665 \ \mathrm{m/s}^2$             gravitational acceleration

$B = \frac{u_{*\mathrm{t}} D_{\mathrm{p}}}{v}$             friction Reynolds number,

initially $B = 1331 (D_{\mathrm{p}}/\mathrm{cm})^{1.56} + 0.38$

$v = 0.157 \cdot 10^{-4} \ \mathrm{m}^2/\mathrm{s}$    kinematic viscosity of air

$z_{\mathrm{o}} = 0.01 \ \mathrm{cm}$             surface roughness length

$z_{\mathrm{os}} = 0.00333 \ \mathrm{cm}$       local roughness length of the uncovered surface

$w$             gravimetric soil moisture in %

$\phi_{\mathrm{clay}}$             clay fraction in %

*Author contributions.* KK conceived the study, implemented the DNN and wrote the manuscript supported by JL. Both authors discussed the results and finalised the article.

*Competing interests.* The authors declare that they have no conflict of interest.

*Acknowledgements.* We acknowledge financial support from the MaxWater Initiative of the Max Planck Society.

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
