# Peer review of "Climate model-informed deep learning of global soil moisture distribution"

_Geoscientific Model Development, 2020_

## Author Comment (AC1)

**Reply to RC1**

We thank the referee for the very helpful comments. We have revised the manuscript accordingly, below please find our replies to the individual comments.

> The paper applies a deep neural network to build a relationship between 18 predictors (simulations of rain, surface temperature and humidity, location, seasonality, root depth etc.) and a predictand (soil moisture down to about 5 cm). The simulation data was produced by a global atmospheric chemistry-climate model EMAC nudged to reanalysis data. The predictand's reference data was the ESA CCI Soil Moisture product.
> The motivation for the application of a neural network was to replace EMAC's soil moisture parameterization with a better one in a mineral dust emission parameterization. The study shall be seen as a proof of concept (line 195). Yes, it is, but a few issues should be clarified.
> The application has very dry areas in its focus. I have in mind that the soil moisture satellite product is especially uncertain in these areas. This should be discussed a bit. The trained prediction is most uncertain in the most interesting regions (Fig. 4: Sahara, Gobi Desert etc.). Why? Quality of the satellite reference or a training period of only 8 years?

In the driest central desert regions, the soil moisture is not very relevant for the dust emissions because the RHS of Eq. (1) is equal or close to unity. It is most relevant in semi-arid regions at the interface of desert and non-desert regions, which is dry enough to allow dust emissions but with enough moisture to have an effect. Therefore the uncertainty in the very dry regions is of limited concern in this context. Fig. S1 in the new supplement shows an example time series in a dry grid cell on the Arabian Peninsula, where comparison and training with the satellite data are challenging because retrievals are both rare and inaccurate. Fig. S2 shows the time series in a grid cell further north in Mesopotamia where the soil moisture levels are more relevant for dust emissions. Here, the observations and predictions are more reasonable and show good agreement. Other applications than dust emissions may require and extra treatment for the central desert regions. This is now mentioned in the discussion of Fig. 4 in section 3.

> The DNN is built with 512 units and four hidden layers. This parameter selection should be motivated a bit. Of more concern is the DNN performance. With location and seasonality as predictors, I expect a high correlation between prediction and reference soil moisture. What is the benefit of using meteorology/climate simulation in the prediction?

We added some motivation for the DNN dimensions in Section 3, but neither a strict rule can be applied, nor did we perform a systematic optimisation of these paramaters, therefore other choices might work even better. In section 3, we also added a discussion of the importance of the meterological predictors which substantially enhance the temporal correlation and allow the predictions to respond to climatic changes.

> Finally, it would be helpful to have short discussions on the applicability of the chosen approach in a changing climate and an alternative DNN training of EMAC parameters (avoiding two parametrizations predicting soil moisture).

We address the applicability in a changing climate in the new discussion on the importance of the meteorological predictors. Regarding the alternative training, it is true that strictly speaking our approach involves two soil moisture parametrisations, the DNN and indirectly the original EMAC parametrisation which is used to produce the training data. To reduce the effect of the latter, the training could be repeated after generating new training data with an EMAC simulation that uses the DNN. But we are not sure whether this is advantageous at this stage.

---

## Author Comment (AC2)

**Reply to RC2**

We thank the referee for the constructive review. We have addressed the comments with a revised manuscript including a new supplement and the point-by-point reply below.

> This study explores DNN-based soil moisture predictions that are trained on remote sensing data and forced by a climate model, which are used to replace low-quality soil moisture predictions in that model for an improved prediction of mineral dust emissions. Overall, the topic is relevant, the methodology novel and sound, and the manuscript concise, well structured, and was a pleasure to read. I just have a few concerns and suggestions which I hope the authors will consider before publication.
> -) To me, hardware developments do not seem to be the most pressing issue and motivation for the presented study. I would recommend to start the introduction outright with the more relevant issue, i.e., the low-quality internal soil moisture predictions of the climate model which could potentially be replaced by a DNN-based module.

The performance and hardware developments are not the most pressing issue for the soil moisture example, where the immediate benefit is a more realistic representation of surface soil moisture. However, we believe the challenge of adapting climate and atmospheric chemistry models to current and future hardware is serious enough to justify the short digression on this aspect. Building on the soil moisture example, computational more demanding model components can be addressed to achieve more significant performance gains.

> -) Maybe a better way to achieve independence between training and test data than doing a standard 80/20 splitting could be to take the characteristics of the ESA CCI SM product into account, that is, splitting the data at a time where there is a change in the underlying satellite instruments that are being merged. Or perhaps better yet: training the model on the "active-only" ESA CCI SM data set, and evaluating it on the "passive-only" data set (but still in a different period of course). That's probably too much to ask for this paper, but perhaps something to keep in mind for future studies.

Taking advantage of the details of the dataset is a very useful proposal for future refinements. So far, we deliberately chose a "naive" approach by just using the most complete data set as is without manually adding additional knowledge about the data (which might not be available in other applications). The split between training and test data was used to obtain more than one year of test data while keeping as much training data as possible. Additionally, we required the training and test data to be in chronological order so that DNN predictions in the test period represent forecasts. For optimal results, future refinements could certainly exploit detailed characteristics of the satellite data. We added the motivation of the data split in section 3.

> -) What motivates the selection of the predictor variables? Are they taken from another study, or did you try out different combinations and evaluated variable importance? A little elaboration on that would be helpful.

The predictors where chosen based on the availability in the simulation output on the one hand

and physical relevance on the other hand. We tested using time lags of more than one week without obtaining a clear improvement of the validation performance. However, since we did not systematically test all possible predictor combinations, better choices are certainly possible. We now elaborate on the predictor choice in section 3.

> -) On a related note: I am a bit concerned that the DNN predicts mostly a spatial and temporal climatology. The correlations of >0.9 are arguably unusually high for typical soil moisture data sets. This argument could be settled, for example, by showing anomaly correlations as well. As for the temporal correlations: The values of ~0.5 indeed appear to be more realistic, but I don't understand the reason for not showing the acutal values in Figure 4. A discussion on how these temporal correlations compare to values typically found for other data sets could also be beneficial. For example, Figure 7 in Dorigo et al. (2017) shows correlation coefficients of modeled versus ESA CCI SM soil moisture, which may serve as a good benchmark for putting the attained values into perspective. Another way to appease the reader may come from an elaboration on the selection of predicture variables, as noted above... For example, how would the predictions look if merely coordinates and the time of the year would be provided? Or if they would be omitted? Also, the time series shown in Fig. 5 do not really convince me. For example, it is pointed to "irregular features" in, for example, October 2014. But to me it seems that there are distinct dry-downs in the prediction in both October 2014 and October 2015, while this dry-down is only visible in the observation in 2014.

The high overall correlation coefficient is dominated by the realistic spatial distribution and can almost be achieved by a climatology, but the temporal correlation is the more challenging contribution and good results are only obtained including the meteorological predictors (Figs. S4 and S5 in the new supplement). We added a discussion of the importance of meteorological vs coordinate/time predictors in section 3 and updated Fig. 4 to show the actual correlation coefficient. The features of the time series in Fig. 5 mentioned in the text are magnified in Fig. S3 in the new supplement.

> -) Having said that, for the application shown, it seems that this may not be relevant at all. The improvements in dust emission predictions shown in Figure 6 are quite remarkable, and it seems that the DNN-predicted values, even it they would be only a climatology with possibly a little short-term variability signal added on top, are doing already much better than the model-internal soil moisture representation. The taken approach of validating soil moisture predictions using a downstream application is a great way to show their actual utility, which is much more useful than the more common approach of simply looking at rather meaninglesss correlation values alone.

Describing the DNN predictions as "climatology with added short-term variability" is not entirely wrong and would indeed be sufficient for many applications. But the added variability is based on meteorology where, e.g, decreased humidity results in lower soil moisture. Therefore, persistent meteorological anomalies result in persistent soil moisture changes (Figs. S6 to S8 of the new supplement) so that also the long-term variability goes beyond a pure climatology. Besides, already the construction of a globally consistent soil moisture climatology based on

satellite data is not trivial, but the DNN obtains the climatological contribution automatically during training.

> -) I'd change the title of Sec. 4 to singular since only a single application is shown.

We propose two independent applications: replacing soil moisture parametrisations in climate models (on which we elaborate using the example of dust emission schemes) and reprocessing observations.

> -) I recommend to add a note of caution to the argument that the DNN could be used to fill gaps in the remotely sensed data. For example, tropical rain forest are masked out entirely in the ESA CCI SM. I am not sure if the DNN is able to properly learn the relation between the predictor variables and soil moisture in such a distinct regime if it is not represented in the training data set at all. A similar argument can probably made for the winter. If soil moisture cannot be retrieved because the soil is frozen or covered with snow, then I would not expect a DNN to properly turn precipitation, which is most likely in the form of snowfall, into accurate predictions of soil moisture.

We have added a note of caution to section 4.

---

## Author Comment (AC3)

**Reply to RC3**

We thank the referee for the valuable comments which contributed to an improved manuscript. In the following please find the point-by-point reply.

> General Comments:
> This paper presents a deep neural network model that predicts global surface soil moisture from precipitation, temperature, and humidity outputs from a climate model. The model was trained on daily satellite retrievals of soil moisture. The authors suggest two uses for the model: 1) to provide modeled soil moisture inputs for related applications, and 2) to fill missing values in satellite retrievals. The authors demonstrate an application by simulating threshold surface friction velocity for mineral dust emission in the Arabian Peninsula and Mesopotamia.
> I found this paper to be rigorous, complete, and convincing. While I have some questions, I think the overall quality is very good. The conclusions are well-founded, and the future research questions are well discussed.
> This paper could use some help from an English language editor. Some sections of the paper are very well written, and some have grammatical and language flaws. Even so, the paper is easy to read and understand.

Language copy-editing will be applied during production.

> Specific Comments:
> The authors present a simulation of threshold surface friction velocity for mineral dust emissions in the Arabian Peninsula and Mesopotamia as an application of the model. They simulate the threshold friction velocity using both observed and DNN-modeled soil moisture with good agreement. This leads me to ask: why use the DNN here at all? Why not just use the observations directly?

The observations have pixels with missing data, Figure 6 reduces their effect by considering the regional mean (and linearly interpolating over the remaining gaps), but a direct use of the observation in a dust simulation would require a proper gap-filling strategy. Secondly, the DNN can be used not only for nudged simulations of periods where observations are available, but also for free running simulations. We mention these motivations in the introduction.

> The authors say that Figure 6 shows "The results based on the observed and predicted soil moisture show good agreement and a strong seasonal cycle". More detail should be given here. The model appears to overpredict the threshold surface friction velocity a bit in the summer. Then, "whereas the result based on the EMAC soil water has little variability" – they could also compare the DNN to EMAC soil water directly.

We added a discussion of the slightly too high summertime thresholds. Another direct comparison of the EMAC and DNN soil moisture would add limited information, because Figs. 5 and Fig. 6 already demonstrate that the deviations are quite substantial. A major reason for these deviations is that the EMAC soil moisture also represents water in deeper soil layers, whereas

the DNN was trained on the surface soil moisture, which is the relevant variable for surface processes. One purpose of Fig. 6 is to compare the threshold estimate that has been available in EMAC simulations with the new estimate based on the DNN which we propose to use instead. The advantage of a better representation of the moisture at the surface by the DNN is now emphasized.

> The authors should provide more information about the model selection criteria they used for the DNN input variables. They used a set of 18 input variables, all of which are intuitive. However, it would be interesting to see which of these input variables drive the predictive power of the model. This might be a particularly interesting question for the mineral dust application: what are the most important drivers of soil moisture in the Arabian Peninsula and Mesopotamia and what does this mean for vulnerability to dust storms? In regions where the temporal correlation is weaker, are the $\cos(2 pi t/a)$ and $\sin(2 pi\ t/a)$ terms dominating to impose the observed seasonal cycle?

We have added the motivation for the choice of predictors and show the sensitivity of example time series to the different predictors in the supplement (Figs. S6 to S8). While generally the analysis of the relevance of different parameters using DNNs is a promising approach and will certainly be focus of future studies, with the present setup one has to be careful because the predictors are not independent. For example, the results in the Middle East are not sensitive to a precipitation reduction, even though a causal relation surely exists. But the sporadic precipitation events hardly coincide in model and reality and make precipitation an unreliable predictor in this region, which the DNN compensates by making more use of the other predictors.

> Figure 5 shows a time series comparison of predicted and observed soil moisture at a single pixel during the test period. This pixel is located in Germany, where the model is reported to have strong temporal correlation (Fig. 4). What does the time series look like in a pixel with a poorer temporal correlation? What does is look like in a pixel in the poorly correlated region of the Arabian Peninsula?

We added a plot of a time series on the Arabian Peninsula with poor temporal correlation to the supplement (Fig. S1). The sparse and uncertain satellite observations are typical in the driest regions.

> Figure 7 shows the global distribution of observed and predicted volumetric soil moisture on two days in the training period. It would be very interesting to see similar plots for the test period.

With post-processing of observational data in mind, where both training and prediction are performed within the same time period, we present data from within the training period in Fig. 7. Note that even though the DNN is evaluated in the training period, the predictions of interest, i.e., predictions for grid cells without observations, are naturally not part of the training data. Of course the trained DNN can also complete observations in the test period and we provide example distributions in Fig. S9 in the new the supplement. However, the model is expected to perform poorer outside the training period so that this is not recommended to obtain best

results for production data sets.

> Technical Corrections:
> There is quite a bit of model evaluation work in the "Applications" section. In particular, I think that the presentation and some of the discussion of Figures 5 and 7 could be moved up.

We moved the paragraph discussing the DNN result in Fig. 5 (and the new time series plots in the supplement) to the evaluation in section 3. The discussion of Fig. 7 remained in the application section because it illustrates the gap-filling application but does not rigorously compare predictions and observations.